# The Health of Nepali Migrants in India: A Qualitative Study of Lifestyles and Risks

**DOI:** 10.3390/ijerph16193655

**Published:** 2019-09-28

**Authors:** Pramod R. Regmi, Edwin van Teijlingen, Preeti Mahato, Nirmal Aryal, Navnita Jadhav, Padam Simkhada, Quazi Syed Zahiruddin, Abhay Gaidhane

**Affiliations:** 1Faculty of Health and Social Sciences, Bournemouth University, Bournemouth BH1 3LH, UK; evteijlingen@bournemouth.ac.uk (E.v.T.); pmahato@bournemouth.ac.uk (P.M.); naryal@bournemouth.ac.uk (N.A.); 2Department of Community Medicine, Datta Meghe Institute of Medical Sciences, Wardha 442001, India; justnavnitaj2907@gmail.com (N.J.); zahirquazi@gmail.com (Q.S.Z.); abhaygaidhane@gmail.com (A.G.); 3Public Health Institute, Liverpool John Moores University, Liverpool L2 2QP, UK; p.p.simkhada@ljmu.ac.uk

**Keywords:** Migration, lifestyle, pre-departure training, Nepali migrants, risk behaviour

## Abstract

*Background*: Most health research on Nepali migrant workers in India is on sexual health, whilst work, lifestyle and health care access issues are under-researched. *Methods*: The qualitative study was carried out in two cities of Maharashtra State in 2017. Twelve focus group discussions (FGDs) and five in-depth interviews were conducted with Nepali male and female migrant workers. Similarly, eight interviews were conducted with stakeholders, mostly representatives of organisations working for Nepali migrants in India using social capital as a theoretical foundation. *Results*: Five main themes emerged from the analysis: (i) accommodation; (ii) lifestyle, networking and risk-taking behaviours; (iii) work environment; (iv) support from local organisations; and (v) health service utilisation. Lack of basic amenities in accommodation, work-related hazards such as lack of safety measures at work or safety training, reluctance of employers to organise treatment for work-related accidents, occupational health issues such as long working hours, high workload, no/limited free time, discrimination by co-workers were identified as key problems. Nepali migrants have limited access to health care facilities due to their inability to prove their identity. Health system of India also discriminates as some treatment is restricted to Indian nationals. The strength of this study is the depth it offers, its limitations includes a lack of generalizability, the latter is a generic issue in such qualitative research. *Conclusion*: This study suggests risks to Nepali migrant workers’ health in India range from accommodation to workplace and from their own precarious lifestyle habit to limited access to health care facilities. We must conduct a quantitative study on a larger population to establish the prevalence of the above mentioned issues and risks. Furthermore, the effectiveness of Nepali migrant support organisations in mitigating these risks needs to be researched.

## 1. Introduction

Migration is a global phenomenon; nearly one-seventh of the world’s population now live in another place than where they were born [1]. The number of international migrants continues to increase; from 173 million people in 2000 to 258 million in 2017 and more than half (150.3 million) are migrant workers [2]. Migration can have major development implications at the individual level but also at the national and global level. 

Migration-related health risks are widely recognised. Recent bibliometric analysis of global migration health research [3] documents a large body of work around migration and health. Another umbrella review about migrant workers health identified various risks including infectious diseases, cardio-metabolic diseases and risk factors, injuries, respiratory diseases, sexual risks, substance misuse and malaria [4]. The Sustainable Development Goals (SDGs) include several targets that mention migration largely within non-health goals (e.g., 4b, 5.2, 8.7, 8.8, 10.7, 10c, 16.2, 17.18), recognising the wider impact of migration [5].

Nepal is a growing supplier of migrant workers. An estimated 3.5 million Nepali are working abroad; primarily in India, Malaysia and the Middle East [6]. Due to limited employment opportunities in the country, many Nepali consider migration as a livelihood strategy. For example, in the fiscal year 2016/2017, Nepali migrant workers sent over US$6.1 billion in remittances which was 26.3% of the country’s total gross domestic product (GDP) [7]. 

Applying theory in studies on migration requires making a number of choices. First, there is the choice of underpinning academic discipline. Migration studies have used a range of theories originating in Economics, International Relations, Social Geography or Sociology [8]. Then there is the choice of ‘level’ of analysis as studies can focus on the macro level (e.g., nation labour markets), the meso level (e.g., social integration in a certain locality) or the micro level (e.g., coping with stress of migration). A further choice may have to be made around focusing on migrants’ sending countries, the receiving countries, others covering both home and host countries. 

Our study based in the host country India, uses the sociological theory of social capital [9]. Social capital can be defined as ‘networks together with shared norms, values and understandings that facilitate co-operation within or among groups’ [10] (p. 41). Social capital includes things like relationships with friend or relatives and people in one’s community and mutual financial or social support [11]. Linking to the latter, our study operates at the meso and micro-levels, that is, the everyday lives of Nepali migrant workers and communities in India. Garip [9] associates the social capital of migrants with information or assistance that they get through their social ties with other migrants who came before them, that is, shared knowledge and understandings linked with expectations of mutual trust and support among people who are part of the same or similar community.

While educated Nepali and those with high economic status migrate to Europe and America, many rural poor, illiterate and unskilled Nepali youths travel to India and Malaysia and the Middle East for work [12]. As India and Nepal have open borders for their citizens and labour permits are not required, cross-border migration to India remains largely undocumented. Due to the lack of proper reporting system, reliable information on cross-border mobility is not available. Different studies have estimated that the number of Nepali in India ranges from 0.5 to 3 million [13]. However, it is estimated that about 1 million Nepali work in India as a temporary migrant [12]. 

Past studies among Nepali migrants in India were predominantly among men and focusing on sexual risk behaviours, for example, reporting high rates of: (a) unprotected sex (33%, often with sex workers [14]; (b) HIV (8%) and (c) syphilis (22%) [15]. Recent rounds of the Integrated Biological and Behavioural Surveillance (IBBS) surveys show lower prevalence of HIV among returnee migrants to India, for example, 0.3% (2015) in Western Hill Districts and 0.6% (2015); 0.4% (2017) in Mid and Far Western Districts and 0.3% (2018) in Eastern Districts of Nepal [16]. National HIV report shows that 17.5% (*n* = 32,747) of ever reported HIV cases in the country as of July 2018 are migrants or spouse/partners of migrants [17] signalling this subgroup is at a higher risk of HIV infection than the general population, as acknowledged by the National HIV Strategic Plan 2016−2021 [18]. Low literacy levels, age at first migration abroad, peer influence, alcohol consumption, living alone abroad, low use of condom, having a sex partner abroad are frequently reported risk factors for Nepali migrants [19,20,21].

Recent studies among Nepali male migrants in the countries of Gulf Cooperation Council (GCC) countries document health vulnerabilities such as anxiety, depression, tuberculosis, accidents and injury, headache and suicide attempts [22]. Similarly, in a recent study with Nepali female migrant workers in the GCC, a quarter reported various health problems. For example, Nepali female migrant workers who were working for unlimited periods of time, changing one’s work place, illiterate, severely maltreated or tortured in the workplace, not paid on time and who had domestic problems were more likely to report health problems [23].

Although there is some evidence on working and living conditions, lifestyles and health and well-being of Nepali migrants in GCC countries and Malaysia [21,22,23,24,25,26], relevant information on cross-border migrants to India is extremely sparse. The experiences of labour migrants to India could be significantly different from those to the GCC and Malaysia because cross-border migration to India is mostly seasonal and circular. Also, they are usually poor, less educated and more likely to work in the informal sector with less protection of labour rights and high risk of exploitation which is likely to impact negatively upon their health and wellbeing. Qualitative research is therefore needed to: explore issues such as (a) accommodation and working environments in the context of health vulnerabilities; (b) lifestyles affecting their health; and (c) use of and access to health care service amongst Nepali migrants in India.

## 2. Methods and Materials

This qualitative study comprises focus group discussions (FGDs) and interviews [27,28]. In early 2017, we carried out twelve FGDs, six each with Nepali male and female migrants in Mumbai and Nagpur, two cities of Maharashtra State of India. The number of participants in the FGDs ranged between five to eight persons. Additionally, five in-depth interviews with participants who did not want to share views in a group setting were carried out. We also interviewed eight key informants (KIIs) with relevant stakeholders, mostly representative working for Nepali migrants in India. Table 1 presents characteristics of our FGD participants.

We had two main approaches to recruiting, namely (a) through local organisations working for Nepali migrants; and (b) with the help of participants/their network, commonly known as snowball sampling [29]. All FGDs were facilitated in a mutually agreed place by an experienced same sex researcher. With the prior permission from the participants, the FGDs and interviews were audio recorded. Most FGDs lasted between one to two hours whereas interviews took between 30 min to one hour. 

Together with key stakeholders the research team drafted discussion guides in the form of questions to facilitate our FGDs and interviews. They were pretested [30] with one FGD with Nepali migrants in Nagpur. The discussion guides included issues such as lifestyle, living condition, risk taking behaviour such as visiting sex workers, utilisation of health care services, work environment and quality of life in India. The FGD guide was used in all FGDs as a starting point and the interview guide was adjusted for each KII depending on their background. The KII guideline focused on questions around migrants’ health and wellbeing issues including support they received from organisations working for migrant community. 

A local researcher [=fifth author] transcribed and translated the FGDs and interviews into English. Three Nepali-speaking authors independently reviewed the transcription and translation. Any disagreements were discussed within the research team for the most appropriate translation. Each transcript had a cover note describing the FGD/interview setting, how the discussion had established, any differences to other interviews, particular incidents, environments and a reflection on the issues raised in the session. Transcriptions were organised through NVivo Ver. 12 (QSR International Pty Ltd, Melbourne, Australia) [31]. Three authors analysed all transcripts and three further authors acted as second coders. Any differences between the coders were discussed in the team until consensus was reached. A thematic approach was performed for data analysis [32]. Relevant quotes are presented to illustrate the key themes. The consolidated criteria for reporting qualitative studies (COREQ) checklist was followed to report the FGDs and interview data [33].

## 3. Ethical Consideration

The present study was approved by Bournemouth University’s Research Ethics Committee (Ref: 13022, approval date: 11.11.2016) and the ethical review board of Datta Meghe Institute of Medical Sciences, India (Ref: DMIMS/IEC/2016-17/4069, approval date 28.09.16). All study procedures were designed to protect participants’ privacy, ensuring anonymous and voluntary participation. Through a participant information sheet in Nepali, participants were provided with information about the study, their voluntary participation, confidentiality, risk and benefits to them, the complaint procedure and so forth. We sought written informed consent [34] from participants prior to the FGDs and interviews. Appropriate travel costs were reimbursed to the participants.

## 4. Results

Five main themes emerged from the analysis of four qualitative data sets: separate FGDs with males and females, in-depth interviews with participants and the KIIs. Our themes were: (i) accommodation; (ii) lifestyles, networking and risk-taking behaviours; (iii) work environment; (iv) support from local organisation; and, (v) health service utilisation. These are discussed below and relevant quotes from participants are presented as illustrations. Table 2 provides a very basic quantitative overview of the qualitative findings. It does not reflect how often the themes were mentioned in an individual data set nor whether theme was regarded as positive, negative or both. In true qualitative style themes following the table do not refer to numbers but to less quantitative descriptions of ‘some,’ ‘few’ or ‘most’ interviewees and/or participants in FGDs. 

### 4.1. Accommodation in India

When asked about accommodation in India, most participants acknowledged that it is very difficult to find accommodation without help of friends, close or distant relatives or acquaintance who are already living there. Generally, they share a room offered by friends. However, some had lived on the street when they first came, as they did not know anyone or did not find a job for some time after coming to India. After newcomers are settled at work or after living for a few weeks or months in temporary accommodation, they gradually find their own rented place with the help from friends and relatives.
“*See, when a person comes first, they stay in a relative’s room or house for a month or so. Then they shift to rented rooms.*”(FGD, Female, I)

Some commented that accommodation might be available from employers, particularly for those working 24-h shifts, such as security guards or hotel workers.
“*Now in our building as we are working here, we have got accommodation here... if someone works outside, then he has to rent a room elsewhere.*”(FGD, Male, V)

Educated Nepali, especially those well established in their job, may possess their own house. Some participants commented that some Nepali enjoy better health and a better life abroad as the following quote on well-paid migrant workers in good employment illustrates:
“*Those who have good jobs and good salary, they are even settled here.*” (FGD, Female, III)

Many problems were reported about rented accommodation or accommodation provided by employers. There was a belief among participants that migrants have to pay higher rent than local Indians for similar standard rooms. Our study found that many Nepali migrants had shared accommodation in a single crammed room, particularly when they just arrive in India or depended on accommodation provided by employers. Some of the rented accommodation did not even offer basic facilities such as toilets and drinking water. Some told us they had to pay extra for using water at their rented place. Participants stated that they had to use public toilets nearby or other places where they are sitting in a queue for such facilities, as exemplified by this quote:
“*No, nothing sir, no drinking water, no water in toilet, we go quite a distance for the latrine. No one is here to help Nepali people. They can’t do anything.*”(FGD, Male, VI)

Only a few participants reported that facilities were good at their rented place and commented that this depended very much on the goodwill of the owner of the house. Although some landlords provided more/better facilities, others were more exploitative and asked Nepali migrant workers to pay extra for basic facilities such as water. 

### 4.2. Lifestyles, Networking and Risk Taking Behaviours 

Most participants accepted that Nepali migrants generally eat healthy meals but most of them were not knowledgeable about nutritious or balanced diet.
“*We do not know what is nutritious; we just eat daal [=lentil soup], rice, subji [=curry] and all.*” (FGD, Female, V)

Common meals include rice, lentils and vegetables, as many would have been used to at home in Nepal. Many said that they also eat meat and fruit but this depends on the financial capacity of the family. Interestingly, many said that meat is eaten only at the weekend:
“*Problems in having a nutritional diet...That’s the problem of poverty. That is how...though they wish to eat good food, due to problems of earnings in cheap labour they cannot eat good food.*”(Key informant interview, VIII)
“*Those who have money they can afford meat or fruit regularly. We have to work hard just to eat rice and curry.*” (In-depth interview, IV)

Participants thought that fast foods are not popular among Nepali migrants. However, new migrants are tempted to eat street and junk food. Many also said that Indian people use a lot of spices and salt in their food and it takes time to adjust their taste to Indian food.
“*People here take more spicy and salty food but we Nepali take less spicy food. So they give us sample food to eat so that we can prepare food accordingly.*”(FGD, Female, I)

Our FGDs found that most of Nepali migrants are physically inactive during out of the work time. Lack of time, work tiredness and usually physically active nature of the work are factors that discourage them for regular physical activity.
“*We don’t get time out of work schedule, when will we exercise? We start working since morning to evening. Some may be doing exercise. But most female migrants don’t get time for it [=exercise].*” (FGD, Female, IV)

One interviewee argued that there was enough physical activity at work, in often very demanding physical jobs:
“*Yes, we do a lot. Exercise happens during work [laughs]!*” (In-depth interview, II)

Participants reported several barriers even they wish to walk, jog or run, including the risk of being hit by motor vehicles and streets without lights in the evening.
“*We do not have the electricity light next to the house, after taking food I want to take a walk but there are no streetlights.*”(FGD, Female, V)

However, one of our KIIs informed that those working in non-physical office-type jobs do engage in physical activities daily:
“*Regularly? Yes. Morning walk. Many brothers living around here do so. I also walk when if I have time, otherwise I exercise at home.*”(Key informant interview, V)

Phones and social media were the popular means of communication among Nepali migrants. However, for many the cost of the internet was a barrier to using social media.
“*Yes! we use Facebook sometimes. New people meet on Facebook. All are connected with WhatsApp.*”(In-depth Interview, II)

Interestingly, few participants commented that dating was common through social media and that there were some online marriage bureaus which helped young Nepali to find their life partners.
“*Like he has opened a marriage bureau. So he helps to get information about bride or groom and meet each other if marriages are planned.*”(FGD, Female, IV)

Participants also acknowledged the importance of networking and communication to find jobs or to learn tasks or to find places to live with the help of friends and relatives.
“*See, there is this lady in my neighbourhood I know. I went with her two or three times. She showed me how to work. Slowly with her help I learned the job and now I am used to the work.*”(FGD, Female, I)

Many participants mentioned that they watch movies, television or go out or play with friends in their free time. They seemed to organise cultural events to celebrate Nepali festivals. Whilst many appeared to be involved in some kind of entertainment, few did not have enough time for this. The main reason they quoted was the lack of time as they worked long hours and returned home after work very late.
“*See all Nepali work at day and night as well, if they meet on the way so we interact with each other, we all are labours, so don’t have timetable to hang out with friends and such things [=cultural events].*”(FGD, Male, I)

We found that drinking alcohol and smoking were common among Nepali migrants. Smokeless tobacco such as *Gutkha* or *Kharra* is very popular among young migrants and some argued that female migrants also equally drink. Some commented that few Nepali drink so much that they end up sleeping on the street. 

Most participants were not aware of anybody taking illegal drugs whilst some thought very few Nepali are involved in such kind of activity. Whereas, only one key informant specifically pointed out that it was very common and there was a problem of drug addiction:
“*Lots of people do use drugs. The drug suppliers run big rackets. These drug dealers have political contacts.*” (Key informant interview, I)

Participants accepted that Nepali male or female migrants engage in extramarital affairs, for example, one female participant explained:
“*Nepali people do marriage at an early age, so their wife works whole day and get very thin physically. Their wife loses facial lustre due to working too hard. So, husband looks for fresh face and gets attracted to local girls. Finally, they give everything to that girl.*”(FGD, Female, VI)

Our participants claimed that those who live alone may also visit sex workers. None of our participants shared their own experience of visiting sex workers or having extramarital affairs. They said that they would not reveal if they had gone to sex workers.

### 4.3. Work Environment

Participants spoke about many work-related problems: low salary, high workload, long working hours, accidents at work, deaths occurring at workplace, lack of safety training provided, unsupportive work environment, health problems due to bad working environment, difficult policies causing problems, no holidays, late payment, no payment for overtime work and so on, for example:
“*We have to work for 24 h. They [=business owner] know how much they spend a day. Why don’t they think about us that we also have to wake up early in the morning? We also have children and families.*”(FGD, Female, III)

Lack of work-related training was common, with newly arrived labour migrants learning on the job, rather than being offered formal training prior to starting work:
“*No! We didn’t get any training before starting the work. Those who are working there from the beginning, they teach us.*”(In-depth interview, V)

Some participants claimed that their poor health was due to their working environment. They argued that no personal protective equipment was provided for working in high-risk environments, leading to many health problems. Participants raised concern around the rising incidence of non-communicable diseases (NCDs) among Nepali in India, which they thought was being caused by high level of pollution in India’s big cities:
“*There is this hot oven, big ovens and because of that there is lot of dust and smoke. So, it causes us breathing difficulty.*”(FGD, Female, II)

Others focused on the problem of pesticide-treated food and the lack of fresh food:
“*Here food has so many chemicals. I came here from Delhi and the situation was terrible there, we don’t get fresh vegetables.*”(FGD, Female, III)

Some specifically commented on the high incidence of mosquito-borne diseases such as malaria and dengue in India. Both male and female participants agreed that most Nepali migrants suffer from stress and poor mental health due to their high workload and/or poor working environment. Our participants experienced some kind of discrimination and harassment mostly at work and sometimes also in their communities. They said that Nepali migrants are teased or treated badly by local people due to their facial appearance.
“*they call small girls kanchi [=young] and kancha [=young] to boys. Elder people call kanchi kancha after them on the street, they get irritated and then they cry.*”(FGD, Female, II)

The key informants agreed that Nepali migrants face a lot of discrimination in India. They are discriminated by employers who often do not give compensation to people who got injured or died due to an accident at work. However, there was a belief among some participants that those Nepali who have good jobs are satisfied as they experience no discrimination at work. A similar view was expressed in the following quote:
“*No, No, Nepali don’t face any problems. Those who love us they call us Chinese and Assamese.*”(In-depth interview, I)

### 4.4. Support from Social Organisations 

Key informants knew of social organisations in different cities in India working to improve the welfare of Nepali migrants, although only a few of our FGDs participants were aware of such charities. Key informants said the roles of these organisations include helping injured (or victims) migrants or their families to get necessary compensations, developing peer networks and unity among Nepali, organising sports or cultural events and celebrate festivals, for example:
“*There are 52 to 55 organisations working for Nepali people. So, these organisations arrange events for them. Nepali people go to cultural functions or sport programs like volleyball and cricket tournaments. Some people go to picnic with groups.*”(Key informant interview, I)
“*We help all the people who are from Nepal and give them money also. We deal with accident cases … we had the Sion hospital burn cases. We try to give them financial help but we have limited resources but we do a little bit for them.*”(Key informant interview, IV)

FGD participants noted many legal and administration problems faced by Nepali migrants as India has difficult rules and policies which can cause problems. Migrant workers in our study thought that these charities should work together with officials in India for migrants’ welfare:
“*We don’t have any proof [=identification] so how we will live here? They will ask for proof to open bank account, for boarding they ask for proof belonging to India, that is far point, they even ask for ID for renting rooms here, now you tell me what we should do? Nepali organisations should take these agenda on board and help us.*”(FGD, Male, III) 

### 4.5. Health Service Utilisation

Participants believed they could get better health treatment in India than in Nepal. However, few mentioned that it is difficult to access health facilities due to lack of transport, the need to have an identification card for using healthcare services and the lack of facilities at government hospitals, for example:
“*They don’t get vehicle easily. They don’t have enough savings to send patient to private hospital. In such condition we go to a government hospital.*”(FGD, Male, VI)

The more positive comments mentioned the relatively good quality of care in Indian government hospitals:
“*We get good care here. We go to nearby government hospital if we need care. In private the health care cost is very high, so we prefer government hospital.*”(FGD, Female, I)

Many acknowledged that there is no discrimination in health facilities. However, a few disagreed that large hospitals provide free health care for some diseases such as cancer treatment to Indians only but not to Nepali migrants. Our KII interviewee claimed that Nepali organisations support poor Nepali immigrants with health-related issues. However, participants surprisingly said that there are no organisations working for migrants’ health. When asked whether they prefer to go to Nepal for treatment if they are ill, most participants said no. Interestingly, they prefer to go Nepal when they need palliative care or Ayurvedic treatment. 

## 5. Discussion

To the best of our knowledge, this is a first study of its kind which explores Nepali migrants’ lifestyles, working environments and their health care services utilisation in India. India has long been a major destination for Nepali workers because of its proximity, established networks, low cost of migration and the open border between the two countries. 

Finding accommodation seems to be easier for those with pre-existing links of friends or relative. This phenomenon of friendship or family relations suggests that those with more social capital had a lower risk of ending up sleeping on the street [9,35]. Studies in Indonesia and Thailand showed the people were more likely to migrate for work abroad if their families had higher social capital resources [9,35]. Our study reported several problems with existing accommodation, including the lack of basic facilities in rented rooms which may be partly due to their low affordability. However, there was a suggestion of exploitation as Nepali migrant workers could end up paying higher rents than their Indian counterparts for the same standard of accommodation. Accommodation provided by the employer was usually sub-optimal, often crammed and lacking basic amenities. The poor living conditions of Nepali migrant workers may make them more vulnerable to infectious diseases such as tuberculosis, malaria or dengue. Research participants reported eating healthy food and, usually, avoiding junk food. However, in line with the findings from other lifestyle studies on Nepali migrants [36], alcohol and tobacco use were common among both male and female Nepali migrants to India. Although participants did not want to elaborate more on sexual practice in India, they did report incidents of extra-marital affairs and visiting sex workers. A study to investigate sexual health practices among Nepali migrants in Maharashtra, India found loneliness and alcohol consumption as reasons for seeking services of sex workers [37]. Whilst Dalit labourers who had migrated to India for economic reasons were more likely to be involved with female sex workers due to peer influences, being unmarried, alcohol use and low-priced sex [20]. 

Similarly, there are multiple problems related to workplace, working environment, co-workers and employers. Participants reported long working hours without or limited weekly/annual leave, lack of safety measures and safety training, work-related accidents and no medical treatment, maltreatment by employers, mental stress due to high workload as the key work-related problems in India. Discrimination of Nepali migrants was also reported by Samuels [38], mostly by co-workers who limit their interaction and exploit them. Nepali guards, for example, face discrimination at work in the form of verbal abuse and denial of services by residents of buildings they guard. Previous studies have reported problems faced by labour migrants including non-payment of wages, physical violence or accusation of theft, difficulty in finding employment, working for long hours and so forth [39]. Domestic workers are at risk of physical and psychological violence by their employers [39]. The report from Asia Foundation found that if women were poorly informed about their job abroad they were more likely to end up working in the informal sector, that is maids in private houses [40]. Thus, migrant workers in the destination countries are more likely to face poor working condition including inadequate salary, lack of social security and protection against abuses and exploitation. Some are even subjected to physical and sexual violence which affects their physical and mental wellbeing [21,41,42]. 

Migration increases the risk of ill health and fatal diseases among migrants as they may take more risks or do not have proper access to healthcare. Our participants mentioned poor access to hospitals, the need for identification documents for treatment and the quality of available service with better treatment to be found in private hospitals. This is in line with other studies in different geographical setting. For example, a study of Nepali and Bangladeshi migrants in India found difficulty in accessing health care facilities due to requirements of having an identity card to get government health facilities, which in turn compels them to use expensive private hospital facilities or delay treatment until they reach home by self-medicating themselves [38]. It is argued that migrants in India suffer from many frequent illnesses due to: (a) lack of health awareness; (b) living in dirty squatters; (c) a lack of basic facilities; or (d) poor hygiene [39]. Discrimination and harassment of migrants may be partly responsible for their reluctance to seeking health care in the destination country and sometimes even at home after their return to Nepal [43]. Many Nepali migrants in India are poorly literate and hence their level of health literacy is also very low. The role of Nepali migrant organisations in India can help to overcome legal and administrative hurdles and to facilitate health care access and treatment for migrant workers. Migrant workers from low-income countries often have to rely on informal networks as a way of minimizing their risks [44].

Despite Nepal being a conservative country where sexuality or other sensitive issues are not openly discussed, most participants actively engaged during the FGDs and interviews. However, we also acknowledge that some of our FGDs were pre-existing groups (i.e., participants know each other already), therefore they might not have shared some sensitive issues (e.g., visiting sex workers, drug misuse) openly with the fear that other member of the group would mention this in their communities. To increase the range of voices we also offered individual interviews to migrants who did not want to speak in a FGD. 

The strength of this study is the depth it offers, its limitations includes a lack of generalizability, the latter is a generic issue in such qualitative research. Therefore, to improve generalizability our next step is to seek funding for a large-scale quantitative survey. This survey will need an appropriate random sample of the Nepali migrant worker’s population to determine the prevalence and severity of the reported problems around their health and lifestyles issues in several key states of India. 

## 6. Conclusions

Nepali migrant workers in India face many challenges. Their low socio-economic status is often reflected in the poor quality of their accommodation and workplace, putting them further at risk of various diseases. Furthermore, discrimination faced by these migrants at workplace, place of residence or while utilising heath care services, adds to the already deprived state. This study also adds to our knowledge of Nepali migrant workers and the need for public health awareness raising in both destination and origin country. The latter should include messages on limiting alcohol and tobacco use and practising safer sex in India. There is a need to offer support and advice on a wide range of health and well-being issues to Nepali migrants in India, especially for those lacking social capital.

## 7. Further Recommendation for Action

The Government of Nepal has implemented a mandatory pre-departure orientation programme to aspiring international migrant workers other than the India. It may not be possible immediately to implement such programme to India-bound Nepali migrant workers due to the open border. However, we strongly recommend a systematic and regular orientation programme in areas of Nepal with high levels of labour migration to India. Such programme may cover issues like common diseases and risk factors in India and ways of preventing them, lifestyle practices (mainly around alcohol and tobacco consumption and safer sex), finding a job and accommodation, health care facilities in India or organisations to solicit support in India. A few migrant related non-governmental organisations (NGOs) conduct health awareness and education programme but this is sporadic and hugely dependent on the project duration. The provincial governments of Nepal (mainly Province 6 and 7) should make this priority in their policy and programme as large number of their population aspire to work in India. In additions, employers and NGOs working for migrants should offer more and better support and advice, thus improving migrants’ social capital. Finally, further quantitative research is needed in India to (1) determine the prevalence of health risks across the total Nepali migrant population; and (2) measure effectiveness of our suggested interventions.

## Figures and Tables

**Table 1 ijerph-16-03655-t001:** Characteristics of focus group participants.

Socio-Demographic Characteristics	Male (*N* = 40)	Female (*N* = 38)
Age		
19–29 years	12 (30%)	15 (40%)
30–39 years	11 (28%)	10 (26%)
40–49 years	13 (33%)	10 (26%)
50 years and above	4 (10%)	3 (8%)
Education		
Literate	4 (10%)	13 (34%)
Primary	11 (28%)	4 (11%)
Lower secondary	21 (53%)	19 (50%)
Secondary	1 (3%)	2 (53%)
Higher secondary	3 (8%)	-
Occupation of participants		
Labourer	6 (15%)	6 (16%)
Security/Watchman	25 (63%)	-
Driver	1 (3%)	-
Cook	3 (8%)	-
Work supervisor	1 (3%)	-
Domestic worker/Cleaner	2 (5%)	32 (84%)
Waiter	2 (5%)	-
Marital status		
Unmarried	5 (13%)	1 (2.6)
Married	35 (88%)	36 (95%)
Widow/widower	-	1 (3%)
Years lived in India		
Up to 2 years	11 (28%)	10 (26%)
3 to 5 years	7 (18%)	9 (24)
6 years and more	22 (55%)	19 (50%)

**Table 2 ijerph-16-03655-t002:** Overview of key themes by migrants and key informants.

Key Themes	Migrants (*n* = 17) *	KIIs (*n* = 8)
Accommodation		
- support friends/family etc.	17	8
- quality of accommodation/facilities	17	3
- discrimination/paying higher rent	13	4
- related to type of job	11	7
Lifestyle, networking and risk-taking behaviours		
- food	17	6
- physical activity	16	8
- social media/networking	15	6
- extra-marital relationships	10	3
- alcohol and smoking	17	8
- other risk-taking behaviour (visit sex workers/drugs)	9	4
Work environment		
- unfair treatment at work (low salary, not timely paid, holiday issues etc.)	10	7
- accidents and injury at work	17	5
- work related training/personal protection equipment	13	4
- impact on health due to work environment	12	2
- discrimination at work	16	3
Support from organisations		
- awareness about support organisation	10	5
- social activities/cultural programmes	4	6
Health service utilisation in India		
- access to health care	17	8
- quality of health services	13	7
- barrier to access/discrimination at health care centres	16	8
- support from local organisation on health issues	11	4

* data sets, not number of people as it includes 12 FGDs and five individual interviews with migrants. Note: Figures in this table should not be used in terms of percentages as numbers reflect the number of datasets (i.e., FGDs or interviews) that mention a particular theme].

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
