# Peer review of "The Health of Nepali Migrants in India: A Qualitative Study of Lifestyles and Risks"

_ijerph, 2019, doi:10.3390/ijerph16193655_

Round 1

Reviewer 1 Report

MAIN COMMENT

Well written paper and easy to read. Yet, the qualitative nature of the methodology is problematic. Reject in its current form, but authors may be invited to submit a new paper considering the main suggestion of the reviewer.

SPECIFIC COMMENTS:

The two main messages of the abstract are: 1. Nepali migrant workers have several problems, including health risks, and 2. There is a need for more quantitative research. The first message is common sense because, as explained in the text, the migrants are informal due to open borders. One may question in how far we need a qualitative study to prove that this is problematic. The second message is that there is a need for more quantitative research. Why was this quantitative part of the study not done before submitting the paper for publication? The qualitative study conducted is useful to identify the different problem areas so that in the next step of the quantitative analysis the authors may also identify the severity of the problems, which in turn may lead to well-documented  recommendations to government, NGOs and other actors. The results section now reads like: “most participants acknowledge”, “generally they share a room”, “however, some participants said …”, “etc”.. One wonders in how far the qualitative questions could not have been analyzed in a more quantitative manner. With a slightly different methodology the authors could have quantified how often certain statements were made. If studying Nepali migrants, there may also be a need to construct a control group of Indian workers doing the same work. This to assure that the problems identified are indeed linked to their migrant status, or, to the contrary, that this is linked to the type of work and that recommendations are applicable not only for the migrants.

SUGGESTION:

There may be two options: 1. The authors conduct a new quantitative study and use this study to orient the questionnaires, methodology and sample size required to arrive at significant findings and recommendations. 2. The authors have a second look at their database and review in how far they can still analyze how often certain statements were made and whether this may still produce significant findings and recommendations. 

Author Response

Response: Thank you so much for your concerns and suggestions. We have revised our background (introduction) section to justify why this qualitative research is needed. We agree with you that there are some studies/quantitative data on migrant’s health; however, we argue that these are mostly based on returnee Nepali migrants from Middle East (Gulf countries) and Malaysia. India is still a major destination (about 37% Nepali prefer to go to India) for many Nepali migrants. Migrants going to India are usually poor, less educated and more likely to work in the informal sector with less protection of labour rights and high risk of exploitation. Moreover, the experiences of migrants to India could be significantly different from those to Gulf and Malaysia because cross-border migration to India is mostly seasonal and circular.

Most studies with Nepali migrants to India are about their sexual behaviors/HIV and STI prevalence. The Government of Nepal also routinely carries out integrated behavioural and surveillance survey (IBBS) with male returnee migrants to India. However, until recently, their lifestyles, and other health and wellbeing issues are largely ignored. Our qualitative study is therefore needed to: explore issues such as a) accommodation and working environments in the context of health vulnerabilities; b) lifestyles affecting their health; and c) use of and access to health care service amongst Nepali migrants in India.” We have a notion that our study highlights the need for a quantitative survey to determine the prevalence and severity of the reported problems around health and lifestyles issues of Nepali migrants in India.

Some of the paragraphs, we have added, revised to justify the need of this study are:

“Recent studies among Nepali male migrants in the countries of Gulf Cooperation Council (GCC) document health vulnerabilities such as anxiety, depression, tuberculosis, accidents and injury, headache, and suicide attempts [18]. Similarly, in a recent study with female migrant workers in GCC, a quarter of respondents reported various health problems. For example, women who were working for unlimited periods of time, changing one’s work place, illiterate, severely maltreated or tortured in the workplace, not paid on time, and who had domestic problems were more likely to report health problems [19].”

“Although there is some evidence on working and living conditions, lifestyles, and health and well-being of Nepali migrants in GCC and Malaysia [17-22], relevant information of cross-border migrants to India is extremely sparse. The experiences of migrants to India could be significantly different from those to GCC and Malaysia because cross-border migration to India is mostly seasonal and circular. Also, they are usually poor, less educated and more likely to work in the informal sector with less protection of labour rights and high risk of exploitation.”

“This is likely to impact upon their health and well-being negatively. Qualitative research is therefore needed to: explore issues such as a) accommodation and working environments in the context of health vulnerabilities; b) lifestyles affecting their health; and c) use of and access to health care service amongst Nepali migrants in India.”

Reviewer 2 Report

General comments:

Abstract is rather long –perhaps try to be a bit more concise. Everything already seems to be in the abstract, which is good in a way, but that may give an excuse to half-baked readers to not read the entire article.

The results/findings section contains a lot of interesting empirical data, it was a great read.

Written English is clear and mostly grammatically correct. However, many sentences would benefit from stylistic improvements as they give the impression that they were written a little too casually.

One or two examples can be found on lines 177-178 or 187. These sentences are alright, but really are not up to academic standard. There are many more sentences that are grammatically correct, but have similar issues. I understand that clarity is more important that unnecessary jargon, but do try to formulate your sentences in a better academic writing style.

Introduction:

Normally, I would expect the introduction to include a clear statement of the authors’ main argument(s) and findings, as well as brief outline of the paper to help the reader navigate the paper more easily. Right now, the introduction does state at the very end that this is a qualitative study and what it is about, but does not specify if it integrates all the previous studies into this one paper or how it may add to the existing body of knowledge on Nepali migrants’ health (and in what way?).

The literature review was interesting to read, and covers many aspects of Nepali migrants’ health, but how does this paper contribute to knowledge? By the end of the introduction, this should have been stated more evidently.

In other words, your abstract normally reads more like an introduction than your introduction does.

The literature review can be left in the intro or move to a separate section ---it doesn’t really matter, as long as the important bits appear in the introduction.

Update as I reach the end of the paper: you do mention line 354 that “this is the first study of its kind”. Your statement here indicates that you may be contributing to the relevant body of knowledge, but your statement remains far too vague. Please specify why/how/in what ways it may be groundbreaking.

Methodology:

It is mentioned line 98 that some of participants were recruited with the help of other participants, by tapping in their networks (b). I am assuming the authors are referring to snowball sampling. Does this mean that some of the focus groups had participants who knew each other? How would that affect the discussion and data collection, especially when speaking about more delicate matters?

From line 110 onward, you discuss how you analyzed the data collected, and who was in charge of whatever task in the process. This is good for clarity and transparency. However, it took me a few too many seconds to realize that some of the authors are referred to by their initials when explaining what their roles were. I am unsure about whether this is common practice or not, editors should be consulted for this decision.

Results:

I understand that this section mostly presents the results only, and discussions will be in a dedicated section later in the paper. However, some bits remain rather descriptive and the sequence/presentation of the findings don’t really help.

Although the results were interesting to read overall, I would recommend that for stylistic purposes, you included mini transitional phrases between your paragraphs and direct quotations like you did on lines 148-9 and lines 157-8 for instance.

Line 136: you mention “pre-exiting link” --- Why don’t you refer directly to social capital with references to the literature (which is plentiful when it comes to social capital and migration decision making, life after migration, accommodation, etc.).

Line 149: “and hence they are happy in India” --- This might be your interpretation, but do be careful, as the direct quotation from Female III does not necessarily indicate or illustrate in any way that anyone is “happy”. I understand why you may have interpreted it that way, but just be aware that it does read a lot like a shortcut that authors took when interpreting qualitative data, especially because there is not much explanation to support that statement.

I understand that this might be trivial and authors might eventually choose to delete those 7 words, but it is a good practice in general to avoid shortcut statements and be more specific.

Regarding direct quotations from participants that were translated: I understand that this is a difficult task, but the translations seems to have a lot of grammatical mistakes. If they are deliberate, please explain why. If they reflect the participants’ speech, be sure to include “sic” to indicate it. Otherwise, it just looks like a sloppy translation even though a lot of effort has been put into it.

Author Response

General comments:

Thank you for your suggestion. We have condensed the abstract. We have carefully addressed all of your concerns. The whole paper has also undergone proofreading.

Introduction:

We have revised our introduction and have added few more paragraphs to justify the need of this qualitative study. We have also edited our discussion section. For example, we have an opening sentence:

“To the best of our knowledge, this is a first study of its kind which exploring Nepali migrants’ lifestyles, working environments and their health care services utilisation in India.”

Methods:

Response: We have added ‘snowball’ sampling approach in our method section. This now reads as:

"We recruited participants mainly from two approaches: a) support from local organisations working for Nepali migrants; and b) with the help of participants/ their network, commonly known as snowball sampling [25]."

We have discussed about the pre-existing group as one of the limitations. This reads as:

“…However, we also acknowledge that some of our FGDs were a pre-existing group (i.e., participants know each other), therefore they might not have shared some sensitive issues (e.g., visiting sex workers, drug misuse) openly with the fear that other member of the group would mention this in their communities.”

We have also removed initials of the contributors as suggested.

We have also added about the social capital concept in our discussion with examples from Thailand and Indonesia, for example:

“This phenomenon of friendship or family relations suggests that those with more social capital had a lower risk of ending up sleeping on the street [32, 33]. Studies in Indonesia and Thailand showed the people were more like to migrate for work abroad if their families had higher social capital resources [32, 33].”

Reviewer 3 Report

Overall, findings are presented extensively and would be ideal for a general research report. However, as a paper for scholarly publication, it lacks the theoretical basis for framing the issues: it is probably somewhere within the paper but it just doesn't come out clearly! 

Check out some minor editing issues e.g.

Line 19: remove "were" on Results should read: five main themes  emerged

Lines 157-158: "they have to long que"should be "they have to que for long....

Line 218: ..."most participants had no idea of anybody taking illegal drugs : remove the phrase in I have bolded (for emphasis): replace with: most participants were not aware of anybody....

Line 310-311: study by Samuels on discrimination not quite related (or at elast the connection isn't succinctly made). The discussion here is on living conditions which is a factor of income, education and exacerbated by immigration status. This is unlike on Line 325 which relates to working conditions and is more aligned to the discussion on discrimination

Author Response

Thank you so much for your valuable suggestions. They are really useful.

We have now revised the background (introduction) section to justify the need of this qualitative research. India is still a major destination (about 37% Nepali prefer to go to India) for Nepali migrants and migrants going to India are usually poor, less educated and more likely to work in the informal sector with less protection of labour rights and high risk of exploitation. Moreover, the experiences of migrants to India could be significantly different from those to Gulf and Malaysia because cross-border migration to India is mostly seasonal and circular. Most studies with returnee migrants (India) are about their sexual behaviors/HIV and STI prevalence. The Government of Nepal also routinely carry out integrated behavioural and surveillance survey (IBBS) with male returnee migrants to India. However, until recently, their lifestyles, and other health and well-being issues are largely ignored. Our qualitative study is therefore needed to: explore issues such as a) accommodation and working environments in the context of health vulnerabilities; b) lifestyles affecting their health; and c) use of and access to health care service amongst Nepali migrants in India.” We have a notion that our study highlights the need for a quantitative survey to determine the prevalence and severity of the reported problems around health and lifestyles issues of Nepali migrants in India.

The whole paper has now undergone proofreading. We have also added concept of social capital concept in our discussions, for example:

“This phenomenon of friendship or family relations suggests that those with more social capital had a lower risk of ending up sleeping on the street [32, 33]. Studies in Indonesia and Thailand showed the people were more like to migrate for work abroad if their families had higher social capital resources [32, 33].”

Round 2

Reviewer 1 Report

The authors added more information from other papers in the discussion section and they edited the paper, which improved its content. The topic of the position of Nepali migrants in India is important, which justifies, in principle, a publication.

Yet, the authors did not address the main recommendation of this reviewer, which was that for the study to merit publication, it is important that the authors quantify how often certain problems were observed by the respondents so that they can arrive at specific recommendations for the intervention strategies.

The results section now reads like: “most participants acknowledge”, “generally they share a room”, “however, some participants said …”, “etc”.. This suggests a semi-quantitative approach, but without data to support the terms “some”, “general”, “most”.

In the first review, two options were suggested to improve the paper :  1. The authors conduct another quantitative study and use this study to orient the questionnaires, sample size required to arrive at significant findings before resubmitting the paper ; 2. The authors have a second look at their database and review in how far they can still analyze how often certain statement were made and quantify the terms “some”, “general”, “most”, etc.

Author Response

Comment 1: The authors added more information from other papers in the discussion section and they edited the paper, which improved its content. The topic of the position of Nepali migrants in India is important, which justifies, in principle, a publication.

Authors' reply: Thank you so much.

Comment 2: Yet, the authors did not address the main recommendation of this reviewer, which was that for the study to merit publication, it is important that the authors quantify how often certain problems were observed by the respondents so that they can arrive at specific recommendations for the intervention strategies. The results section now reads like: “most participants acknowledge”, “generally they share a room”, “however, some participants said …”, “etc”.. This suggests a semi-quantitative approach, but without data to support the terms “some”, “general”, “most”. In the first review, two options were suggested to improve the paper :  1. The authors conduct another quantitative study and use this study to orient the questionnaires, sample size required to arrive at significant findings before resubmitting the paper ; 2. The authors have a second look at their database and review in how far they can still analyze how often certain statement were made and quantify the terms “some”, “general”, “most”, etc.

Authors’ reply: Thank you for your suggestions. We have further analysed our qualitative datasets and have presented the key findings in a quantitative table. The new Table 2 in the Result section provides a very basic quantitative overview of the qualitative findings.  It does not reflect how often the themes were mentioned in an individual data set nor whether theme was regarded as positive, negative or both. In true qualitative style themes following Table 2 do not refer to numbers but to less quantitative descriptions of ‘some’, ‘few’ or ‘most’ interviewees and/or participants in FGDs. We have a notion that this qualitative study highlights the need for a quantitative survey to determine the prevalence and severity of the reported problems around health and lifestyles issues of Nepali migrants in India.

Reviewer 3 Report

After reviewing the article for the second time, I'm glad that the methodology section is very clear and well documented. However, i am still troubled that although the authors claim that they have added a "social capital concept" in their discussion of results, that is not reflected throughout the paper. I'm still struggling to find what their theoretical/conceptual framework is. It seems to be more of a descriptive report than a paper undergoing peer review for publication.

Author Response

Comment: After reviewing the article for the second time, I'm glad that the methodology section is very clear and well documented. However, i am still troubled that although the authors claim that they have added a "social capital concept" in their discussion of results, that is not reflected throughout the paper. I'm still struggling to find what their theoretical/conceptual framework is. It seems to be more of a descriptive report than a paper undergoing peer review for publication.

Authors' reply:

We have added more texts around theoretical underpinning (=social capital) of our study. For example:

In Abstract:

…. were conducted with stakeholders, mostly representatives of organisations working for Nepali migrants in India using social capital as a theoretical foundation.

In Introduction we have added two paragraphs:

Applying theory in studies on migration requires making a number of choices. First, there is the choice of underpinning academic discipline. Migration studies have used a range of theories originating in Economics, International Relations, Social Geography or Sociology [8]. Then there is the choice of ‘level’ of analysis as studies can focus on the macro level (e.g., nation labour markets), the meso level (e.g., social integration in a certain locality) or the micro level (e.g. coping with stress of migration). A further choice may have to be made around focusing on migrants’ sending countries, the receiving countries, others covering both home and host countries.    

Our study based in the host country India, uses the sociological theory of social capital [9]. Social capital can be defined as ‘networks together with shared norms, values and understandings that facilitate co-operation within or among groups’ [10]. Social capital includes things like relationships with friend or relatives and people in one’s community, and mutual financial or social support [11]. Linking to the latter, our study operates at the meso and micro-levels, i.e. the everyday lives of Nepali migrant workers and communities in India. Garip [9] associates the social capital of migrants with information or assistance that they get through their social ties with other migrants who came before them, i.e., shared knowledge and understandings linked with expectations of mutual trust and support among people who are part of the same or similar community.

In Discussion and Conclusion and Recommendations:

….This phenomenon of friendship or family relations suggests that those with more social capital had a lower risk of ending up sleeping on the street [9, 35]. Studies in Indonesia and Thailand showed the people were more like to migrate for work abroad if their families had higher social capital resources [9, 35].

…There is a need to offer support and advice on a wide range of health and well-being issues to Nepali migrants in India, especially for those lacking social capital.

…In additions, employers and NGOs working for migrants should offer more and better support and advice, thus improving migrants’ social capital.